# Effect of Phosphorus Fertilization on Yield of Chipping Potato Grown on High Legacy Phosphorus Soil

**Yuheng Qiu** [1], **Thioro Fall** [1], **Zhihua Su** [2], **Fernando Bortolozo** [1], **Wendy Mussoline** [3], **Gary England** [4], **David Dinkins** [5], **Kelly Morgan** [6], **Mark Clark** [6] and **Guodong Liu** [1,*]

1   Horticultural Sciences Department, University of Florida/IFAS, Gainesville, FL 32611, USA; yuheng.qiu@ufl.edu (Y.Q.); tfall@ufl.edu (T.F.); fernandorb@ufl.edu (F.B.)
2   Department of Statistics, University of Florida, Gainesville, FL 32611, USA; zhihuasu@stat.ufl.edu
3   Putnam County Extension Office, University of Florida/IFAS, East Palatka, FL 32131, USA; wmussoli@ufl.edu
4   Extension Agent IV Emeritus, University of Florida/IFAS, Deland, FL 32724, USA; gke@ufl.edu
5   Office of the Dean of Extension, Florida Cooperative Extension Service, University of Florida/IFAS, Gainesville, FL 32611, USA; dinkins@ufl.edu
6   Department of Soil and Water Sciences, University of Florida/IFAS, Gainesville, FL 32611, USA; conserv@ufl.edu (K.M.); clarkmw@ufl.edu (M.C.)
*   Correspondence: guodong@ufl.edu; Tel.: +1-(352)273-4814

**Abstract:** Potato (*Solanum tuberosum* L.) has low phosphorus (P) use efficiency as compared with other vegetable crops. This study was conducted at two commercial chipping potato farms (A and B) in Northeast Florida to evaluate different P rates for potato production. Plot size was 0.62 and 0.49 hectares for Farms A and B, respectively. The total trial area was 13.32 hectares per growing season for three consecutive years from 2018 to 2020. The randomized complete block design (RCBD) was employed with four replications per P rate. The chipping potato variety, 'Atlantic' was cultivated with three P rates: 0, 24.5, and 48.9 kg P ha$^{-1}$ in 2018, and 12.2, 24.5, and 48.9 kg P ha$^{-1}$ in 2019 and 2020. The soil of Farm A contained 497 mg kg$^{-1}$ Mehlich-3 extractable P and 946 mg kg$^{-1}$ aluminum (Al), and that of Farm B had 220 mg kg$^{-1}$ Mehlich-3 extractable P and 253 mg kg$^{-1}$ Al. The results showed that a P rate of 48.9 kg P ha$^{-1}$ significantly improved tuber yield as compared to 0 in 2018 or 12.2 kg P ha$^{-1}$ in 2019 and 2020. Application of 48.9 kg P ha$^{-1}$ fertilizer P significantly increased the soil P level in 2018 and 2020, while the application of 24.5 kg P ha$^{-1}$ fertilizer P increased the soil P level significantly in 2018 only. Tubers with 48.9 kg P ha$^{-1}$ showed significantly lower external quality issues than 0 or 12.2 and 24.5 kg P ha$^{-1}$. However, there were no significant differences in specific gravity, internal tuber quality, and tuber size among the different P rates. The tuber yield data show that potato plants grown on soil with high legacy P still require approximately 50 kg ha$^{-1}$ P application for sustainable potato production in the area. This high P requirement results from the combination of high concentrations of active metals (Al and iron (Fe)) and a decrease in pH of one unit in the growing season. New P-fertilization programs with post-plant applications rather than with pre-plant application are urgently needed for minimizing P-immobilization by Al and Fe and improving P-use efficiency for potato production in the state.

**Keywords:** potato production; rootzone phosphorus; best management practices; soil aluminum content; potato tuber quality

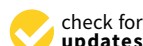



## 1. Introduction

Potato has been long consumed in the human diet since it is rich in starch content and carbohydrate calories [1]. Potato consumption has been expanding in the developing countries [2], and it is one of the most consumed food crops in the U.S. [1] In the past decade, US potato production has almost remained unchanged with a total of 25 million tons per year, valued at approximately $650 million [3]. Idaho and Washington lead the nation in potato production with combined production accounting for nearly half

of the US total. Ranking as 13th in the US, Florida's potato is valued at approximately 150 million US dollars [3]. Potato requires more phosphorus (P) than any other vegetable crop since its undeveloped root systems cannot effectively mobilize insoluble phosphates from the soil [4–6]. As a nutrient resource essential for plant growth and development, non-renewable rock phosphate reserves may be depleted by 2050 [7,8]. Aluminum (Al) and iron (Fe) bound P is abundant in many soils making the nutrient unavailable to crop plants in acidic soils such as in Northeast Florida [8]. Therefore, phosphorus is often the most limiting nutrient for plant growth in acidic soils, constraining crop production on 30–40% of global arable land [8]. For a crop with a high P requirement such as potato, excessive P application can also cause water pollution—eutrophication—a serious environmental issue that deprives certain waterbodies of dissolved oxygen and can lead to algae blooms, that generate toxins harmful to human and animal health [9].

In Northeast Florida, more than 7000 hectares of potatoes are currently grown in Flagler, Putnam, and St. Johns counties, commonly known as the "Tri-County Agricultural Area (TCAA)". Excessive P has accumulated in these soils because of continuous crop cultivation since the 1890s, with P fertilizer application since the 1920s [10]. Liao et al. reported that the soils are rich in Mehlich-3 extractable P ranging from 200 to 600 mg $kg^{-1}$ [10], which is substantially greater than the soil Mehlich-3 level (45 mg $kg^{-1}$) for where no P fertilizer would be recommended [11]. If available to crop plants, this P range is more than enough for most vegetable crops but may be insufficient for potato production because potato has a shallow and less developed root system, and the soil has elevated Mehlich-3 extractable Al. Other factors can also influence soil P bioavailability. For example, organic amendments have been shown to increase bioavailable P for potatoes [12].

The potato yield response to P fertilization is a conundrum. Tuber yield is expected to be more responsive to fertilizer P when a soil test P level is low [5]. However, some studies found that in conditions of limiting soil P, P fertilizer application resulted in a higher total yield with more but smaller tubers [13,14]. Related research also found that P fertilizer led to more but smaller tubers per plant and thus resulted in a higher total yield [15]. Soil pH plays an important role in affecting tuber yield response to P fertilizer. In low-pH soils, adequate liming must be practiced before P fertilizer application to minimize combined fixation by soil Al and/or Fe that renders P unavailable for plant uptake [16]. Recent studies had found that split P applications through drip irrigation can significantly increase the P-use efficiency and potato productivity [17]. Soil texture also plays a role in affecting P-use efficiency in potato plants. Higher yields were observed in the potato plants applied with a higher P fertilization rate, and the trend was found stronger in the clayey soil as compared to the sandy soil [18]. Instead of using the conventional Mehlich-3 P soil test, studies had found that the Mehlich-3 concentration percentage between P and Al ($(P-Al)_{M-III}$) can be a better criterion for making both environmentally and agronomically valid P recommendations for potato production in P-enriched soils [19,20]. The soil P fertility and environmental risk were divided into three groups (low, medium, and high) with three critical $(P-Al)_{M-III}$ values: 4–8%, 8–15%, and >15% [20].

To better understand potato yield response to P fertilizer in conditions of excessive Mehlich-3 soil P, a three-year potato trial was conducted from 2018 to 2020. The purpose of this study was to evaluate the potato yield response to three P rates (0, 24.5, and 48.9 kg P $ha^{-1}$) when grown in soil with high legacy P. This research aimed to test if a higher P fertilization rate still increases potato productivity when potatoes are grown on P rich soil. Parameters including tuber yields, specific gravity, internal and external tuber quality, and tuber sizes were measured to verify the hypothesis: P fertilization will significantly increase tuber yield of chipping potato plants grown on soil with high-legacy P due to high concentrations of active Al and Fe and pH decreases in the growing season. The objectives of this study were to: (1) evaluate different P rates for chipping potato production in large-scale plots on commercial potato farms in Northeast Florida; (2) better understand the effects of P application on specific gravity, internal and external quality, and size of potato tubers; and (3) compare the economic return of phosphorus fertilization.

## 2. Materials and Methods

### 2.1. Experimental Design

A three-year (2018–2020) experiment was conducted on two commercial farms located in the TCAA area using 'Atlantic' cultivar (chipping potato) with seepage irrigation. The randomized complete block design (RCBD) was used with four replications of each P application rate. The plot size was 0.62 hectares and 0.49 hectares for Farms A and B, respectively (the total trial area was 13.26 hectares per year). There were three treatments including three different P rates, which were the same for the two farms. As shown in Figure 1A, P rates were 0, 24.5, and 48.9 kg P ha$^{-1}$ during 2018. As shown in Figure 1B, P rates were slightly adjusted to 12.2, 24.5, 48.9 kg P ha$^{-1}$. The P rate of 12.2 kg P ha$^{-1}$ was applied for the control in 2019 and 2020 based on the University of Florida/IFAS phosphate fertilizer recommendations for potato production (https://edis.ifas.ufl.edu/publication/CV002 (accessed on 23 March 2022)) [21] (Figure 2). The P source was ordinary superphosphate and was applied by UF/IFAS employees with a 4-row applicator two to four weeks before planting for all three years. The plot layout at both farms was all the same to see the accumulative effects of P applications. The 12.2 kg P ha$^{-1}$ were applied by the growers at emergence during Years 2 and 3 of the trial. Potato seed pieces were planted at Farm A on 8 February 2018, on 16 January 2019, and on 20 January 2020. At Farm B, the corresponding planting dates were 22 January 2018, 14 January 2019, and 20 January 2020. All irrigation water was from the same well at either farm. There was a fallow row per 16 rows for irrigation purposes. Seepage irrigation was used to keep the water table between 0.46 and 0.61 m. The total rainfall and evapotranspiration (ET) at Farm A were 196 and 259 mm in 2018, 248 and 288 mm in 2019, and 92 and 269 mm in 2020. Those at Farm B were 294 and 281 mm in 2018, 248 and 289 mm in 2019, and 92 and 265 in 2020. The daily ranges of rainfall and ET were 0 to 55 mm and 0 to 5 mm in 2018; 0 to 42 mm and 0 to 5 mm; and 0 to 14 mm and 0 to 4 mm in 2020 for both farms. The monthly soil temperatures at the two farms from January 2018 through June 2020 are listed in Table S1.

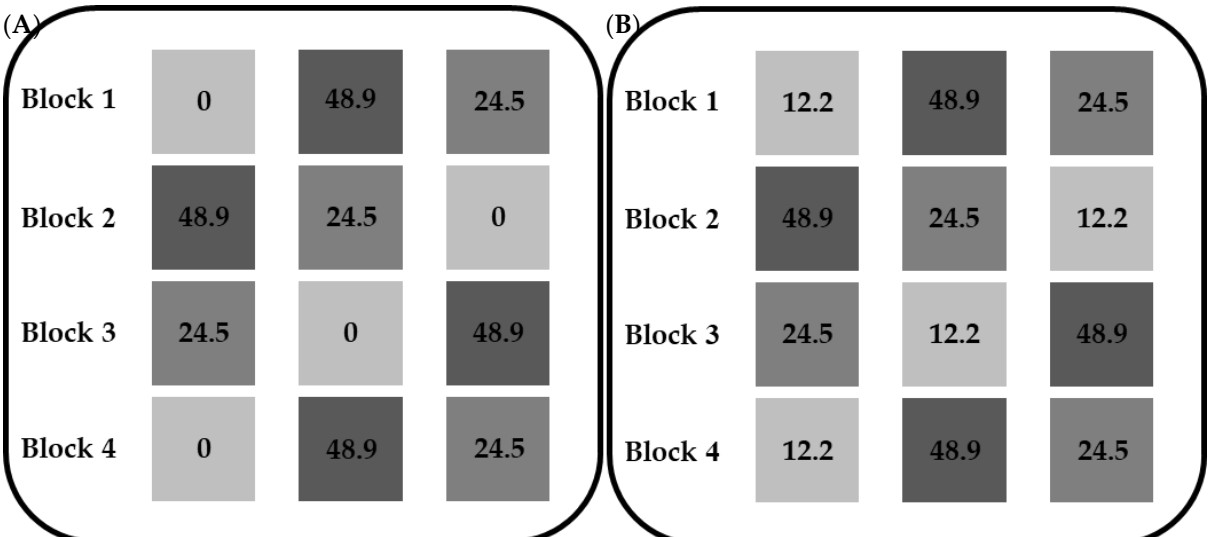

**Figure 1.** P trial layout on Farms (**A**,**B**) in 2018 (**A**) and in 2019 and 2020 (**B**). Numbers refer to fertilizer P rates (kg P ha$^{-1}$). When the soil temperature is low at planting 12.2 kg P ha$^{-1}$ is applied as starter fertilizer according to the UF/IFAS recommendation for P management for potato production (Table 4 at https://edis.ifas.ufl.edu/publication/CV002 (accessed on 23 March 2022)). The plot arrangements were all the same in each of the three years.

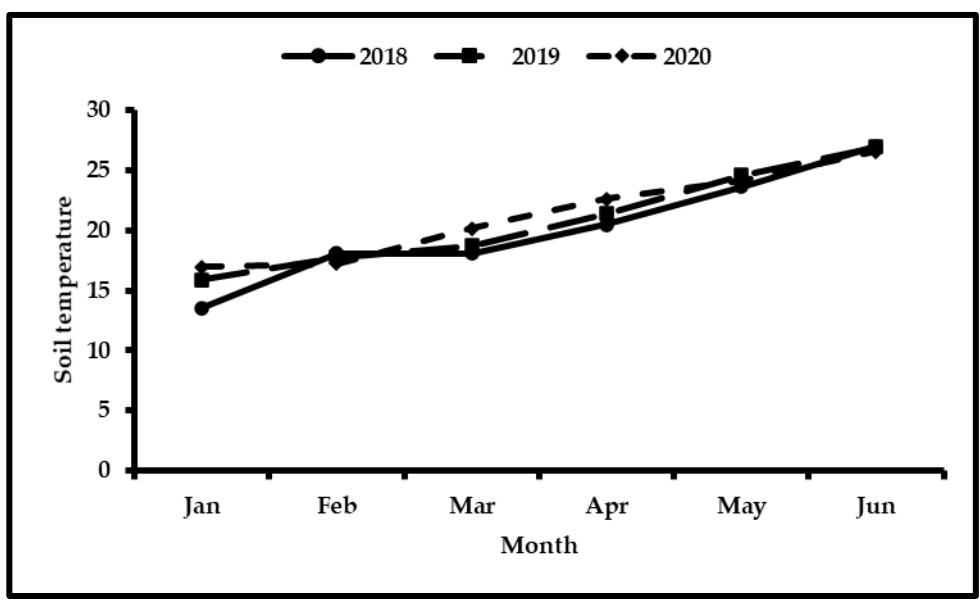

**Figure 2.** Soil temperature at potato growing season from January to June for the years 2018, 2019, and 2020.

*2.2. Farm Managements*

Farm A: the soil contained 497 mg P kg$^{-1}$, 946 mg Al kg$^{-1}$, and 187 Fe mg kg$^{-1}$ in soil tests using Mehlich-3 extraction (Table 1). Two split-applications with 280 kg N ha$^{-1}$ in total were made per growing season: Urea Ammonium Nitrate (UAN 32-0-0) was the only N source applied during Year 1, and a combination of UAN 32-0-0 and polyblend 10–34–0 was applied during Years 2 and 3 to provide the additional 12.2 kg P ha$^{-1}$ supplied by the growers. Split N applications were applied with an 8-row applicator on each on 30 January 2018; 29 January 2019; and 31 January 2020, and on 21 February 2018, 20 February 2019, and 22 February 2020. The polyblend of UAN 32-0-0 and 10-34-0 added during Years 2 and 3 was applied during the second of the split N applications. The grower spread 260 kg K ha$^{-1}$ as 0-0-55 blended from half potassium sulfate and half potassium chloride with an 8-row Diversified Products spreader before planting. The grower made 7 applications on a 7-day schedule of 1 quart Dyna Green Potato mix (1.5% chelated magnesium, 3% combined sulfur, 0.15% water-soluble boron, 0.5% chelated iron, 2.5% chelated manganese, 0.02% water-soluble molybdenum, and 1.5% chelated zinc, lignin sulfonate as the chelating agent for all of the above chelated nutrients) and 1 pint of Borsol 10 (10% boron as boric acid). Foliar applications were made with a 16-row sprayer and no gypsum was applied during any of the trial seasons.

**Table 1.** Average soil Mehlich-3 extractable P, Al, and Fe for Farms A and B.

| Farms | Element | Concentration |
|---|---|---|
| | | mg Element kg$^{-1}$ soil |
| Farm A | P | 497 |
| | Al | 946 |
| | Fe | 187 |
| Farm B | P | 220 |
| | Al | 253 |
| | Fe | 312 |

Farm B: the soil contained 220 mg P kg$^{-1}$, 253 mg Al kg$^{-1}$, and 312 mg Fe kg$^{-1}$ in the soil test with the Mehlich-3 extraction (Table 1). Three split applications of both N and K were applied before planting, at emergence, and layby each year at slightly different

rates (Table 2). Nitrogen was applied in three split applications: pre-plant consisted of 50% each of ammonium nitrate and ammonium sulfate and was spread with a 4-row fertilizer band applicator; at emergence and layby during the growing season it consisted of 75% ammonium nitrate and 25% ammonium sulfate and was spread with an 8-row fertilizer band applicator. Potassium was blended from 50% each of potassium sulfate and potassium chloride and spread in three split applications with the same equipment listed above. Nitrogen and potassium application rates were listed in Table 2. The grower made 5 split applications on a 5-week schedule totaling 10 L Potato Formula including 2.24% S, 3.0% Mn, 0.5% Cu, 0.8% Fe, 0.005% Mo, and 0.29% Zn. Foliar applications were with a 16-row sprayer. No gypsum was applied in 2018 but 1120 kg ha$^{-1}$ of gypsum was applied in each of 2019 and 2020.

**Table 2.** Fertilizer (N and K only) rates (kg ha$^{-1}$) at Farm "B" for each of the three years.

| Trial Year | Nutrient | Preplant | Emergence | Layby | Total |
|:---:|:---:|:---:|:---:|:---:|:---:|
| 2018 | N | 56 | 202 | 37 | 295 |
| 2019 | N | 56 | 213 | 90 | 359 |
| 2020 | N | 56 | 202 | 45 | 303 |
| 2018 | K | 168 | 121 | 49 | 339 |
| 2019 | K | 168 | 128 | 54 | 350 |
| 2020 | K | 168 | 121 | 45 | 334 |

*2.3. Soil Mehlich-3 P, Al, and Fe Measurements*

Across the three-year trials, soil samples were collected 6 times to monitor the dynamics of soil extractable P during each of the growing seasons: pre-plant in late December or early January, then biweekly two weeks after planting starting from late January or early February to late April or early May. A Varomorus soil sampler probe with a 53 cm stainless steel tubular T-style handle was used for collecting soil samples. The soil samples were collected from all rows except the outside rows in each of the plots, respectively. Approximately, a subsample was collected approximately every 5 m except 5 m from either end in the plot length of 381.40 m at Farm A and of 301.43 m at Farm B. At Farm A, roughly 70 subsamples were fully mixed into one soil sample while at Farm B, 60 subsamples into one sample. The samples were sent to Waters Agricultural Laboratories, Inc. in Camilla, Georgia for nutrient analyses using Mehlich-3 extractant, including glacial acetic acid, 200 mM; ammonium nitrate, 15 mM; nitric acid, 13 mM; and ethylenediamine tetraacetic acid (EDTA), 1 mM. The soil samples were scooped with an NCR standard 2 g scoop and placed in an extraction cup. An Oxford Dispenser was used to dispense 20 mL of the Mehlich-3 extracting solution into each sample cup for 60 samples dispensed in less than 2 min. The samples were shaken for 5 min using the shaker at 200 rpm. Before planting, soil samples were collected for soil chemical analysis of either farm in spring 2018. Mehlich-3 extractable P, K, Ca, and Mg were 182 ± 19, 104 ± 12, 1140 ± 169, and 152 ± 34 mg ha$^{-1}$ at Farm A and 120 ± 22, 37 ± 15, 742 ± 113, and 65 ± 17 mg ha$^{-1}$ at Farm B. The CEC was 6.3 ± 0.2 meg (100 g)$^{-1}$ at Farm A and 3.6 ± 0.2 meg (100 g)$^{-1}$ at Farm B. For the chemical fixation of P by Al, stoichiometrically, one mole of Al can immobilize one mole of P. Similarly, one mole of iron can fix one mole of phosphorus. Gravimetrically, one kg Al or Fe can immobilize 1.15 or 0.56 kg P.

*2.4. Duration and Tuber Yield Measurements*

Potato tubers were dug out with a 4-row harvester (Advanced Farm Equipment LLC, Vestaburg, MI, USA) on Farm A on 8 May 2018, 7 May 2019, and 29 April 2020, and on Farm B on 6 May 2018, 7 May 2019, and 28 April 2020 (Table 3). Tuber yields were measured by using Fairbanks Scales (Model#: 6610, Woodridge, Chicago, IL, USA) at harvest. The regression of tuber yield on P rate, year of measure, and farm is shown in Figure S1. Two 32-kg bags of potato tubers were randomly collected from all the 16 rows in each of the plots from both farms and were used for measuring tuber external and internal quality at

the packing line at harvest in 2018 and 2019 but this was not done in 2020 because of the COVID-19 pandemic. From the two bags, 40 tubers were randomly selected for specific gravity (SG) measurements [22,23]:

$$SG = \frac{\text{Tuber weight in air}}{\text{Tuber weight in air} - \text{tuber weight in water}}$$

**Table 3.** Potato harvest years, date and tuber yield, internal and external quality measurements.

| Farm | Harvest Year | Harvest Date | Internal/External Quality Measurement |
|------|-------------|-------------|--------------------------------------|
| Farm A | 2018 | 05–08 | Yes |
|  | 2019 | 05–07 | Yes |
|  | 2020 | 04–29 | No |
| Farm B | 2018 | 05–06 | Yes |
|  | 2019 | 05–07 | Yes |
|  | 2020 | 04–28 | No |

### 2.5. Tuber Quality Measurements

Forty potato tubers were randomly collected from each plot and used for external and internal quality measurement and specific gravity evaluation (Table 3). This process is a standard method at the University of Florida/IFAS. As the plot scale was large, the potato samples were randomly collected twice: first, two 32-kg bags of potato tubers from all the 16 rows of each of the plots and then 40 tubers from the two bags to ensure the sample representativeness of all the potatoes tubers from each of the plots. The percentage of potato tubers defects was calculated accordingly. The regression of total cull on P rate, year of measure, and farm is shown in Figure S5.

### 2.6. Economic Return of Phosphorus Applications

Only a basic calculation was done based on (1) the income difference between the control and either 24.5 or 48.9 kg P ha$^{-1}$ according to the potato yield and the potato price ($0.29 kg$^{-1}$) during the project period; (2) Phosphorus fertilizer costs for potato production at either farm; and (3) all the other costs of each of the treatments were assumed equal.

### 2.7. Statistical Analysis

Statistical analysis was performed using R statistical software, version 4.0.2 (R Core Team, 2020), and conducted by multiple linear regression with potato tuber yields response, P rate, the farm, and year of measure as predictors. All predictors were treated as factors. Specifically,

$$
\begin{aligned}
\text{Yield} = \text{intercept} \quad &+ c_1 \times \text{P rate} \left(24.5 \, \text{kg·P·ha}^{-1}\right) \\
&+ c_2 \times \text{P rate} \left(48.9 \, \text{kg·P·ha}^{-1}\right) + c_3 \times \text{Year 2019} \\
&+ c_4 \times \text{Year 2020} + c_5 \times \text{Farm B} + \varepsilon
\end{aligned}
$$

where the intercept represents the expected yield in Farm A using a P rate of 12.2 kg P ha$^{-1}$ in the year 2018, $c_1$, $c_2$, $c_3$, $c_4$, and $c_5$ are coefficients for the corresponding predictors ($c_1$ is the coefficient for the P rate at 24.5 kg P ha$^{-1}$ which represents the difference between the expected yield using a P rate at 24.5 kg P ha$^{-1}$ and a P rate at 12.2 kg P ha$^{-1}$, $c_2$ is the coefficient for a P rate at 48.9 kg P ha$^{-1}$, $c_3$ is the coefficient for the year 2019, $c_4$ is the coefficient for the year 2020, and $c_5$ is the coefficient for Farm B). The symbol $\varepsilon$ represents the variation in yield not explained by the P rate, year, and farm. We use 0.05 as the cut-off point for statistical significance, which means if the p-value of a coefficient is less than 0.05, we infer that the coefficient is significantly different from 0. The alpha value was set as 0.05. The relationship between the extractable P content in the soil with the P rate, the P

level, and other mineral contents before fertilization, was also evaluated by multiple linear regression. An overview of the variables is included in Table 4.

**Table 4.** Descriptive statistics of yield, P rate, year, and farm.

| Farm | A | 36 observations |
|---|---|---|
| | B | 27 observations |
| **P rate** | 12.2 kg P ha$^{-1}$ | 21 observations |
| | 24.5 kg P ha$^{-1}$ | 21 observations |
| | 48.9 kg P ha$^{-1}$ | 21 observations |
| **Year** | 2018 | 18 observations |
| | 2019 | 24 observations |
| | 2020 | 21 observations |
| **Yield** | Min: 24,683; Max: 45,373 Mean: 38,101; Median: 39,490 Std Dev: 5161.4 | 63 observations |

## 3. Results and Discussion

### 3.1. Regression of Yield on P Rate, Year, and Farm

After fitting the multiple linear regression, the results on the regression of yield on the P rate, year, and farm are displayed below in Table 5. Note that the intercept represents the yield for Farm A in the year 2018 with a P rate at 12.2 kg P ha$^{-1}$. The 48.9 kg P ha$^{-1}$ fertilizer P significantly increased tuber yield. The yield also varied from year to year, as well as farms. This regression has an R$^2$ of 0.79 (Table 5).

**Table 5.** Regression of yield on the P rate, year of the measure, and farm.

| Variables | Coefficients | *p*-Value |
|---|---|---|
| Intercept | 34,891.6 | $<2 \times 10^{-16}$ * |
| P rate (24.5 kg P ha$^{-1}$) | 682.0 | 0.319 |
| P rate (48.9 kg P ha$^{-1}$) | 1819.4 | 0.00959 * |
| Year (2019) | 3527.1 | $4.10 \times 10^{-6}$ * |
| Year (2020) | −5434.4 | $2.42 \times 10^{-10}$ * |
| Farm B | −2950.1 | $2.61 \times 10^{-6}$ * |

* Significant at $p < 0.05$.

Maximizing potato tuber yield is always the goal for potato production. Yield maximization ensures potato grower's income. In Northeast Florida, the soil has accumulated much extractable P which is unnecessarily bioavailable for potato plants. The results of this study showed both 24.5 kg P ha$^{-1}$ and 48.9 kg P ha$^{-1}$ fertilizer P rate significantly increases the P level in soil, but only 48.9 kg P ha$^{-1}$ fertilizer P significantly increases potato yield (Table 5). The significant yield increase with 48.9 kg P ha$^{-1}$ indicated that the P level increased by the P rate of 24.5 kg P ha$^{-1}$ was still not necessarily sufficient for potato vines. The significant yield increase by 48.9 kg P ha$^{-1}$ was the profitability of potato production in the area. Obviously, potato requires more P than other crops, 117 kg ha$^{-1}$ is recommended in some regions [24]. Khiari et al. and Benjannet et al. reported that the optimum rates for potato P fertilization ranged from 21 to 105 kg P ha$^{-1}$ when soil fertility is low ((P-Al)$_{M-III}$) <5%) [19,20]. Theoretically, no P fertilization is necessary for potato growing in P saturated soil. However, starter P fertilizer is still applied to increase the yields as potato responds efficiently at the early growing stages [19]. The factors responsible for soil P bioavailability include the 4R's nutrient stewardship, soil pH, and soil temperature. Integrating the above factors may optimize fertilizer P rates and enhance P use efficiency for potato production.

### 3.2. Regression of Soil P Concentration (after Fertilization) on P Rate and Soil Al and Fe Concentrations (before Fertilization)

We performed linear regression with soil P concentration as the response variable. The predictors included P rate, days in the growing season, farm, the pH level, aluminum (Al), phosphorus (P), and iron (Fe) contents before the P fertilization remained in the model (the measurements for Fe and Al were not available for 2018 and 2019). With the data from each year, we fit the following model:

$$
\begin{aligned}
\text{P concentration} \quad &(\text{after fertilization}) \\
&= \text{ intercept} + c1 \times \text{P rate} \left(24.5 \text{ kg·P·ha}^{-1}\right) \\
&\quad + c2 \times \text{P rate} \left(48.9 \text{ kg·P·ha}^{-1}\right) \\
&\quad + c3 \times \text{daysinthegrowingseason} \\
&\quad + c4 \times \text{P concentration (before fertilization)} \\
&\quad + c5 \times \text{Farm B} + c6 \times \text{soilpH} + c7 \times \text{Alconcentration} \\
&\quad + c8 \times \text{Feconcentration} + \varepsilon
\end{aligned}
$$

where the intercepts represent the expected yield in Farm A using a P rate of 12.2 kg P ha$^{-1}$ and other predictors (days in the growing season, P, Al, Fe concentrations, and soil pH) taking values zero, c1 to c8 are coefficients for the corresponding predictors. The symbol $\varepsilon$ represents the variation in yield not explained by the predictors. For years 2018 and 2019, since the measurements of Fe and Al are not available, c7 and c8 along with their *p*-values are not available (as appeared as N/A in Table 6). We removed two outliers from the measurements in 2020. Soil pH, Al, and Fe played a dominant role in affecting P contents as soil P was fixed by Al and Fe in acidic conditions [25,26]. Phosphorus rate and farm were treated as factors and other variables were treated as numerical. The coefficients and *p*-values of these variables are listed below (Table 6). The diagnostic graphs are included in Figures S2–S4.

**Table 6.** Regression of P content on P rate, days in the growing season, farm, soil pH, and nutrient contents before fertilization for the years 2018, 2019, and 2020.

| Variables | Coefficients | | | *p*-Value | | |
|---|---|---|---|---|---|---|
| | 2018 | 2019 | 2020 | 2018 | 2019 | 2020 |
| Intercept | 125.05 | 415.3 | 932.720 | 0.436 | 0.159 | $5.35 \times 10^{-5}$ * |
| P rate (24.5 kg P ha$^{-1}$) | 34.653 | 10.45 | 7.186 | 0.00572 * | 0.498 | 0.527 |
| P rate (48.9 kg P ha$^{-1}$) | 45.969 | 12.72 | 22.996 | 0.000302 * | 0.412 | 0.0365 * |
| Days after planting for individual growth stages $^{z}$ | −0.006053 | 2.397 | 0.396 | 0.982 | $<2 \times 10^{-16}$ * | 0.00956 * |
| P (before fertilization) | −0.214 | 1.392 | 1.387 | 0.276 | $1.32 \times 10^{-11}$ * | $2 \times 10^{-16}$ * |
| Farm B | −97.184 | −117.16 | −282.073 | $1.56 \times 10^{-6}$ * | 0.000256 * | $3.10 \times 10^{-6}$ * |
| Soil pH (before fertilization) | 17.492 | −32.68 | −101.595 | 0.495 | 0.130 | 0.00275 * |
| Al (before fertilization) | N/A | N/A | −0.464 | N/A | N/A | $2.19 \times 10^{-6}$ * |
| Fe (before fertilization) | N/A | N/A | 0.175 | N/A | N/A | 0.134 |

$^{z}$ The individual growth stages including pre-plant, emergence, vegetative, tuber initiation, tuber bulking, and tuber harvest. * Significant at $p < 0.05$.

For the year 2018, both 24.5 kg P ha$^{-1}$ and 48.9 kg P ha$^{-1}$ fertilizer P significantly increased the extractable P level in the soil after its application (Table 6). The regression had an R$^2$ of 0.42. Diagnostic plots for all three regression models showed no outliers or nonlinear trends. The results for the year 2019 exhibited days in all the individual growing seasons, P content before fertilization, and the farm were significant variables (Table 6). The regression had an R$^2$ of 0.81. The results for the year 2020 show that 48.9 kg P ha$^{-1}$ fertilizer P significantly increased the Mehlich-3 extractable P level in the soil after its application. Legacy soil P concentration also played a significant role in soil extractable P [27]. The soil

pH and Al content affected the extractable P level after the fertilization as well (Table 6). This regression had an $R^2$ of 0.92, indicating the regression model captures most of the variation in yield.

### 3.3. Soil pH and Chemical Analysis

The soil pH of Farm A and B were measured throughout the entire growing season. Across the three-year trials, soil pH on both farms was lower than 6.0 during most of the growing season, regardless of fertilization rates, which left these farms prone to P immobilization due to high Al and Fe activity. In Farm B, soil pH was as low as 5 from March to May for the years 2018 and 2019 (Figures 3–5). During this growth stage of tuber initiation and bulking, potato vines need the most P, but aluminum and iron at low soil pH < 5.0 immobilize both fertilizer P and legacy P. Other minor factors such as soil temperature and moisture, should also be considered. Low soil temperature leads to decreased development of potato roots [28], and the P mineralization process will stagnate when the soil is dry and cold [25,29].

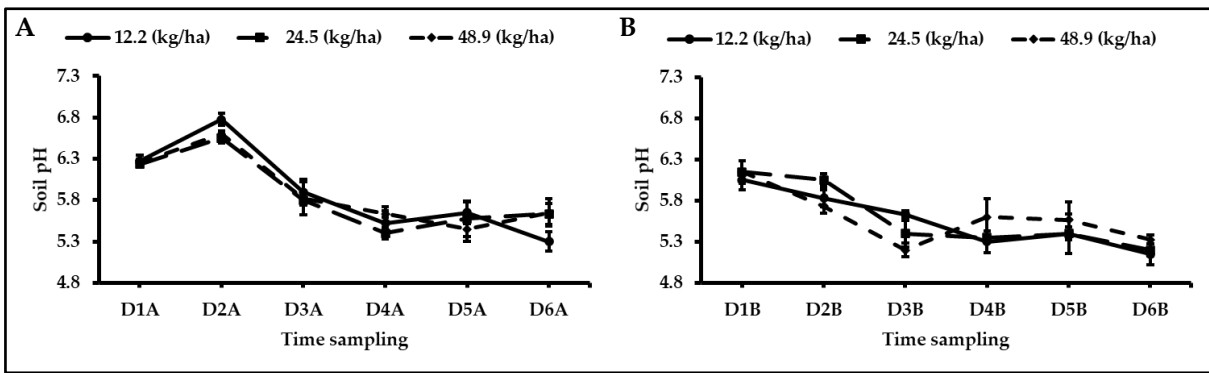

**Figure 3.** Soil pH of two farms (Farm (**A**); Farm (**B**)) with different P fertilizer application rates (0 kg P ha$^{-1}$; 24.5 kg P ha$^{-1}$; and 48.9 kg P ha$^{-1}$) in the year 2018. Each D represents a date: D1A = 4 January 2018; D2A = 22 February 2018; D3A = 7 March 2018; D4A = 23 March 2018; D5A = 12 April 2018; D6A = 26 April 2018; D1B = 12 December 2017; D2B = 5 February 2018; D3B = 7 March 2018; D4B = 23 March 2018; D5B = 12 April 2018; and D6B = 30 April 2018. Error bars exhibit the standard error of the mean from four replicates of the measurements.

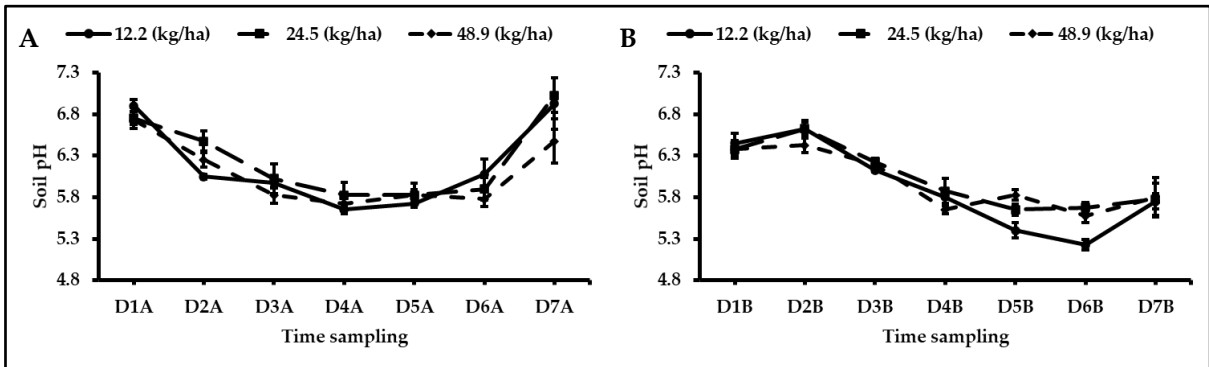

**Figure 4.** Soil pH of two farms (Farm (**A**); Farm (**B**)) with different P fertilizer application rates (12.2 kg P ha$^{-1}$; 24.5 kg P ha$^{-1}$; and 48.9 kg P ha$^{-1}$) in the year 2019. Each D represents a date: D1A = 21 December 2018; D2A = 30 January 2019; D3A = 13 February 2019; D4A = 27 February 2019; D5A = 13 March 2019; D6A = 27 March 2019; D7A = 25 April 2019; D1B = 6 December 2018; D2B = 28 January 2019; D3B = 11 February 2019; D4B = 25 February 2019; D5B = 11 March 2019; D6B = 25 March 2019; and D7B = 22 April 2019. Error bars exhibit the standard error of the mean from four replicates of the measurements.

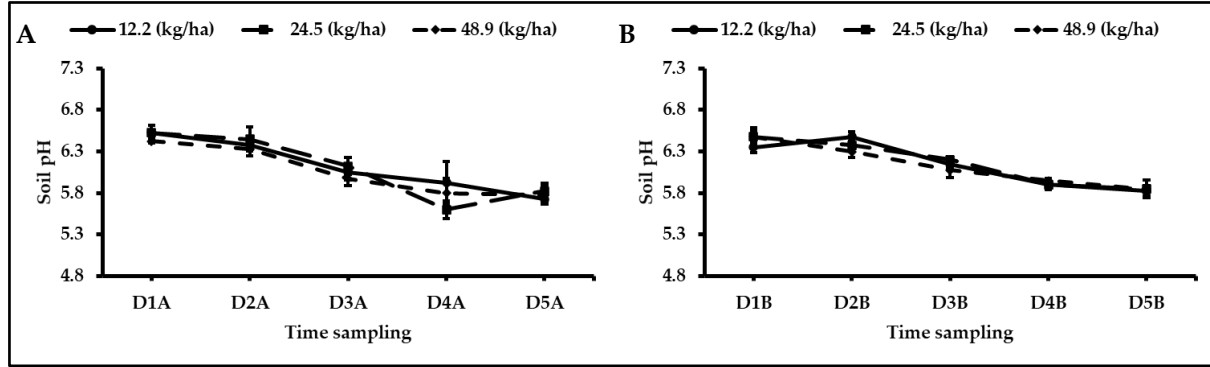

**Figure 5.** Soil pH of two farms (Farm (**A**); Farm (**B**)) with different P fertilizer application rates (12.2 kg P ha$^{-1}$; 24.5 kg P ha$^{-1}$; and 48.9 kg P ha$^{-1}$) in the year 2020. Each D represents a date: D1A = 13 December 2019; D2A = 3 February 2020; D3A = 17 February 2020; D4A = 2 March 2020; D5A = 6 April 2020; D1B = 9 December 2019; D2B = 3 February 2020; D3B = 17 February 2020; D4B = 2 March 2020; and D5B = 6 April 2020. Error bars exhibit the standard error of the mean from four replicates of the measurements.

The soil Mehlich-3 extractable P contents of Farm A and B were measured before the experiment to give the background soil information of each farm (Table 7). The soil Mehlich-3 extractable Al and Fe contents do not change much and were only determined before planting in 2020. One kg ha$^{-1}$ Al and Fe could stoichiometrically combine with 2.63 and 1.27 kg P ha$^{-1}$ in water but the situation can be much different in soil due to the soil chemistry complexity. The P equivalent of Al and Fe were calculated based on the amount of soil established P multiplied by the stoichiometric factor. Therefore, the immobilized P by Al and Fe for Farm A was stoichiometrically up to 5575 kg ha$^{-1}$ and 532 kg ha$^{-1}$, respectively. Farm B had lower established P and Al-bounded P, which was 1130 kg ha$^{-1}$ and 1491 kg ha$^{-1}$, respectively. However, it possibly fixed up to 889 kg ha$^{-1}$ of P due to its higher Fe contents (Table 7).

**Table 7.** Soil Mehlich-3 extractable P, Al, and Fe for Farm A and B in 2020.

| Farms | Element | Concentration (mg Element kg ha$^{-1}$) | P Equivalent [z] (kg P ha$^{-1}$) |
|---|---|---|---|
| | P | 1114 | N/A |
| Farm A | Al | 2121 | 2434 [y] |
| | Fe | 419 | 232 [x] |
| | P | 493 | N/A |
| Farm B | Al | 567 | 651 |
| | Fe | 699 | 388 |

[z] P equivalent refers to the fixed amount of P by the given Al and Fe level. [y] 1 kg ha$^{-1}$ Al can potentially fix 1.15 kg ha$^{-1}$ P (2.63 kg ha$^{-1}$ P$_2$O$_5$) in water but can be different in soil because the complexity of soil chemistry. [x] 1 kg ha$^{-1}$ Fe can potentially fix 0.56 kg ha$^{-1}$ P (1.27 kg ha$^{-1}$ P$_2$O$_5$) in water.

The yield results of the study do not fit with the theory that the soil with high extractable P does not need P application [11]. This discrepancy may be attributed to the high amount of active Al and Fe from the soil parent materials. Stoichiometrically, the active Al and Fe in the soil can immobilize 2666 kg ha$^{-1}$ for Farm A and 1039 kg ha$^{-1}$ for Farm B. These immobilization capacities were 54-fold greater for Farm A and 20-fold greater for Farm B than the greatest P rate used in this study (Table 7). The data contribute a clear understanding of how important sustainable P management for potato production in the area is.

This study showed that soil pH decreased by approximately one pH unit in each of the three growing seasons (Figures 3–5), particularly, in the tuber initiation and bulking

stage when potato plants mostly required P. This pH decrease increased the activity of soil Al and Fe. The soils on both commercial farms contained Al and Fe (Table 7). The metal activity increase resulting from the pH decrease might stoichiometrically immobilize up to 2434 kg ha$^{-1}$ from both fertilizer P and soil legacy P (Table 7). Because of the P immobilization, potato plants require more P fertilizer to sustain their tuber yield production. Soil pH is also critical to make further decisions on other amendments, such as using gypsum. Gypsum amendment also plays a part in tackling soil Al toxicity, as the calcium in gypsum can replace sodium in sodic soils and/or aluminum when soil aluminum is a concern [16]. Organic matter is another crucial factor in controlling P bioavailability in soil, not only by providing the plant with available P via organic matter mineralization but by reducing P retention and adsorption on the soil surface [25,26].

Soil pH plays an important role in P bioavailability and potato yield. Currently, P fertilization is neutralizing the Al and Fe in the acidic soil with a high amount of Al and Fe. It may be easy for us to think of using agricultural lime. However, potato is susceptible to common scab caused by the bacteria *Streptomyces*. This bacterial disease is highly associated with soil pH. This bacterium disease can be suppressed if soil pH is lower than 5.2 [30,31]. Before scab-resistant varieties are commercially available, we need to be very careful about using agricultural lime to adjust soil pH and lower the solubility of Al and Fe.

About P-use efficiency, large root surface areas with fine roots and long root hairs are crucial for nutrient uptake from soil by plants, particularly for acquisition of P [32]. Potato vines require more P fertilizers as compared to other crops due to their shallow root systems with small root surface areas, which cannot extract nutrients and minerals from the soil efficiently [4,6,33]. The optimal soil temperature for potato root development is 20 °C, and root development decreases significantly when the soil temperature is at 15 °C or lower and at 25 °C or higher [28]. However, most potatoes in Florida are planted at the beginning of the year and the growing season is generally from January/February to May/June, which is not optimal for potato root growth at this initial stage of the growing season.

### 3.4. Potato Tuber Quality Measurements

Specific gravity (SG) is a key criterion in measuring the starch content in potatoes (Table 8) [34]. Potato tubers with a SG greater than 1.080 are suitable for processing [35]. The SG for chipping potatoes from Farm A (1.09) was constant for all fertilizer P rates. However, tubers from Farm B had an SG of 1.086 at 24.5 kg P ha$^{-1}$ and 1.084 at 12.2 and 1.083 at 48.9 kg P ha$^{-1}$ (Table 8). No significant difference of specific gravity was observed between the P rates tested. Specific gravity will increase with the P fertilization rate when the soil test P level is low, but additional P fertilization will have little impact on SG when the soil test P level is high [33]. Small-scale potato trials performed in Northeast Florida also found that the tuber SG does not respond to P fertilization in the conditions of excessive soil P concentration [33]. Our results were in accordance with the results of previously published work.

**Table 8.** Specific gravity of 'Atlantic' chipping potato grown on Farm A and B with different P fertilizer applications in the years 2018 and 2019. There was not any statistically significant difference in specific gravity between the P rates.

| Farm | P Rate (kg P ha$^{-1}$) | Specific Gravity | | |
|---|---|---|---|---|
| | | 2018 | 2019 | Average |
| | 12.2 [z] | 1.087 | 1.081 | 1.084 |
| Farm A | 24.5 | 1.086 | 1.083 | 1.085 |
| | 48.9 | 1.087 | 1.082 | 1.085 |
| | 12.2 [z] | 1.082 | 1.085 | 1.084 |
| Farm B | 24.5 | 1.086 | 1.085 | 1.086 |
| | 48.9 | 1.083 | 1.083 | 1.083 |

[z] In 2018 as the first year of the trial, the P rate was zero.

Both the internal and external quality and size of potato tubers were also reported (Tables 9–11). Farm A showed lower issues of both internal and external quality (Tables 9 and 10). The two farms exhibited the same level of potato tuber size (Table 11). There was no statistically significant difference in external quality between 12.2 and 24.5 kg P ha$^{-1}$, but the total cull was significantly lower with 48.9 kg P ha$^{-1}$. The hollow heart was the major internal quality issue, and green skin and growth cracks were the major issues for external quality. All farms and P rates showed a defect rate of lower than 3% for internal quality, while the defect rate of external quality was at a higher range from 10% to 20% (Tables 9 and 10). Most tuber sizes fell into the A1 category (Table 11). Our results were similar to the findings of those small-scale potato trials conducted in Northeast Florida where the tuber quality did not respond to P fertilization from 0 to 74 kg P ha$^{-1}$ at all sites where soil testing of the P level was high [33].

**Table 9.** Internal quality of 'Atlantic' chipping potato grown on Farms A and B with different P fertilizer applications in the years 2018 and 2019.

| Farm | P Rate (kg P ha$^{-1}$) | Internal Quality Issues (% of Total Tuber Yield) | | | | |
|---|---|---|---|---|---|---|
| | | Hollow Heart | Brown Center | Corky Ring Spot | Internal Heat Necrosis | Total |
| Farm A | 12.2 [z] | 1.32 | 0 | 0 | 0.15 | 1.47 |
| | 24.5 | 1.67 | 0 | 0 | 0 | 1.67 |
| | 48.9 | 0.94 | 0 | 0 | 0 | 0.94 |
| Farm B | 12.2 [z] | 0.83 | 0 | 0.50 | 0.83 | 2.17 |
| | 24.5 | 0.78 | 0 | 0.78 | 0 | 1.56 |
| | 48.9 | 1.88 | 0 | 0.16 | 0.16 | 2.19 |

[z] In 2018, the first year of the trial, the P rate was zero.

**Table 10.** External quality of 'Atlantic' chipping potato grown on Farms A and B with different P fertilizer applications in the years 2018 and 2019.

| Farm | P Rate (kg P ha$^{-1}$) | External Quality Issues (% of Tubers Examined) | | | | |
|---|---|---|---|---|---|---|
| | | Green Skin | Growth Cracks | Misshapen | Rotten | Total |
| Farm A | 12.2 [z] | 2.94 [y] | 7.60 | 0.59 | 0.95 | 12.09 |
| | 24.5 | 3.69 | 6.62 | 0.60 | 1.24 | 12.15 |
| | 48.9 | 3.93 | 4.62 | 0.69 | 0.73 | 9.98 |
| Farm B | 12.2 [z] | 7.15 | 5.42 | 1.36 | 2.55 | 16.48 |
| | 24.5 | 8.02 | 4.70 | 1.22 | 2.25 | 16.19 |
| | 48.9 | 5.98 | 3.51 | 0.82 | 3.06 | 13.37 |

[z] In 2018, the first year of the trial, the P rate was zero. [y] Percentage of tubers with quality defects in 40 randomly collected potato tubers.

**Table 11.** Tuber size of 'Atlantic' chipping potato grown on Farms A and B with different P fertilizer applications in the years 2018 and 2019.

| Farm | P Rate (kg P ha$^{-1}$) | Potato Tuber Size (Diameter) | | | | | | |
|---|---|---|---|---|---|---|---|---|
| | | C [y] (12.7–38.1 mm) | B (38.1–47.6 mm) | A1 (47.6–63.5 mm) | A2 (63.5–83.8 mm) | A3 (83.8–101.6 mm) | A4 (>101.6 mm) | Total |
| Farm A | 12.2 [z] | 1.42 | 4.37 | 39.68 | 9.18 | 6.75 | 0.15 | 61.54 |
| | 24.5 | 0.88 | 3.30 | 35.75 | 9.84 | 5.49 | 0.00 | 55.26 |
| | 48.9 | 1.17 | 4.28 | 44.80 | 9.86 | 3.78 | 2.16 | 66.05 |
| Farm B | 12.2 [z] | 1.23 | 4.37 | 33.14 | 11.67 | 12.79 | 0.23 | 63.44 |
| | 24.5 | 0.94 | 3.26 | 31.24 | 13.29 | 14.34 | 0.20 | 63.27 |
| | 48.9 | 0.77 | 2.68 | 30.17 | 13.34 | 16.56 | 0.22 | 63.73 |

[z] In 2018, the first year of the trial, the P rate was zero. [y] C, B, A1, A2, A3, A4 are criteria used to measure the potato tuber size.

The P rate showed significant differences in terms of potato external quality (Table 12). A linear regression model was fit to the data with P rate, year, and farm as predictors and total

cull measure as a response. All predictors are treated as factors. Seven to eight measurements were taken for each farm at each P rate every year, giving a total of 95 measurements. Year, Farm B, and P rate at 48.9 kg P ha$^{-1}$ had significant effects on the total cull, as shown in the table. Specifically, Farm A in the year 2018 had more culls. There was no significant difference between the P rates at 12.2 and 24.5 kg P ha$^{-1}$, but 48.9 kg P ha$^{-1}$ reduced the number of culls (Table 12). The R$^2$ of this regression is only 0.29, suggesting a substantial level of noise (unaccounted for variation) in this regression. However, Year, Farm B, and P rate (48.9 kg P ha$^{-1}$) still showed a significant effect despite the relatively large noise level. Phosphorus plays an important role in affecting potato tuber quality. Higher tuber quality including higher dry matter content has been observed in the potato grown in the soil with higher bioavailable P concentration [36]. Therefore, a high P fertilization rate can lead to a better tuber quality due to the increase of soil bioavailable P concentration.

**Table 12.** Regression of total cull on P rate, year of measure, and farm.

| Variables | Coefficients | *p*-Value |
|---|---|---|
| Intercept | 5.9785 | <2 × 10$^{-16}$ * |
| P rate (24.5 kg P ha$^{-1}$) | −0.0685 | 0.8994 |
| P rate (48.9 kg P ha$^{-1}$) | −1.0998 | 0.0433 * |
| Year (2019) | −1.9754 | 2.15 × 10$^{-5}$ * |
| Farm B | 1.5579 | 6.42 × 10$^{-4}$ * |

* Significant at $p < 0.05$.

*3.5. Economic Analysis*

Potato yield gains and economic returns for all three years were evaluated to give a broad view of the commercial perspective (Tables 13 and 14). The assumed price for an Mg of potatoes is 577 US dollars, and the cost of P fertilizer is 93 US dollars per hectare. However, differences among the P rates on yield were not significant if we only focus on one farm or one year. This non-significance can be attributed to variation due to the large plot size. These plot sizes were 100-fold to 127-fold greater than the regular plot size (49 m$^2$) used in potato studies in the area. For example, if only the 12 measurements (4 for each P rate) for farm A in 2018 were used, there was not enough statistical power to detect the difference among the P rates.

**Table 13.** Potato yield gain and return in Farms A and B.

| Farm | Year | P Rate (kg P ha$^{-1}$) | Yield (Mg ha$^{-1}$) | Yield Gain (kg ha$^{-1}$) | Yield Return per kg P |
|---|---|---|---|---|---|
| Farm A | 2018–2020 | 12.2 | 43.4 | N/A | N/A |
|  |  | 24.5 | 43.8 | 375 | 6.7 |
|  |  | 48.9 | 45.0 | 1561 | 13.9 |
| Farm B | 2018–2020 | 12.2 | 39.0 | N/A | N/A |
|  |  | 24.5 | 40.5 | 1553 | 27.7 |
|  |  | 48.9 | 42.2 | 3236 | 28.9 |

Tuber yield gains were calculated by comparing the yield of different P rates with the P rate of 12.2 kg P ha$^{-1}$, which was set as a reference for the comparison. The calculated yield gains were then divided by the applied P rate to give a potato yield return per kilogram of P. The tuber yields increased with fertilizer P rates, and 24.5 kg P ha$^{-1}$ and 48.9 kg P ha$^{-1}$ generated positive tuber yield returns. Additionally, the efficacies are consistent for these two rates as 48.9 kg P ha$^{-1}$ gave higher returns than 24.5 kg P ha$^{-1}$ (Table 13). P fertilizer cost and potato income for each farm were also calculated. The economic return at each P rate is listed in Table 14. Economic return for 24.5 and 48.9 kg P ha$^{-1}$ ranges from −$11 to $810 and $195 to $1410 per hectare, respectively. Farm B showed a greater return than Farm

A, and 48.9 kg P ha$^{-1}$ gave a more stable economic return than 24.5 kg P ha$^{-1}$ in general (Table 14).

**Table 14.** P fertilizer cost and potato market price at Farms A and B.

| Farm | Year | P Rate (kg P ha$^{-1}$) | Potato Market Price (US Dollars ha$^{-1}$) | Economic Return (US Dollars ha$^{-1}$) |
|---|---|---|---|---|
| Farm A | 2018 | 12.2 | 13,106 | N/A |
| | | 24.5 | 13,188 | 83 |
| | | 48.9 | 13,589 | 484 |
| | 2019 | 12.2 | 13,916 | N/A |
| | | 24.5 | 14,007 | 91 |
| | | 48.9 | 14,172 | 257 |
| | 2020 | 12.2 | 10,447 | N/A |
| | | 24.5 | 10,458 | 11 |
| | | 48.9 | 10,641 | 195 |
| | 2018–2020 | 12.2 | 12,489 | N/A |
| | | 24.5 | 12,551 | 62 |
| | | 48.9 | 12,801 | 312 |
| Farm B | 2018 | 12.2 | 10,440 | N/A |
| | | 24.5 | 10,980 | 540 |
| | | 48.9 | 11,192 | 751 |
| | 2019 | 12.2 | 13,027 | N/A |
| | | 24.5 | 13,015 | −11 |
| | | 48.9 | 13,382 | 355 |
| | 2020 | 12.2 | 9539 | N/A |
| | | 24.5 | 10,349 | 810 |
| | | 48.9 | 10,949 | 1410 |
| | 2018–2020 | 12.2 | 11,380 | N/A |
| | | 24.5 | 11,713 | 333 |
| | | 48.9 | 12,137 | 757 |

The year-over-year analysis demonstrated that both 24.5 kg P ha$^{-1}$ and 48.9 kg P ha$^{-1}$ fertilizer P rate significantly increased the P level in the soil after application in 2018 and 2020 when compared with the 0 or 12.2 kg P ha$^{-1}$ control (Table 6). The results of our study showed 48.9 kg P ha$^{-1}$ fertilizer P significantly increased the potato yield (Table 5), even though the soil Mehlich-3 extractable P levels in these two farms were as high as 220 and 497 mg kg$^{-1}$ and would have resulted in a recommendation of no fertilizer P according to the current University of Florida/IFAS recommendations (Table 1). Additionally, significant reductions were observed in the external quality of potato tubers with 48.9 kg P ha$^{-1}$ fertilizer P (Table 12). Previous studies conducted in Northeast Florida have shown that the potato tuber yields did not respond to a high P fertilization rate when soil testing P level is high [37]. However, our findings would indicate that a high concentration of legacy P in the soil does not necessarily suggest lower P fertilizer application rates; contrarily, tuber yield and external tuber quality continued to increase with the addition of P fertilizer (up to 48.9 kg P ha$^{-1}$). Potential factors that affect potato response to P fertilizer such as cultivars and soil texture should also be included in future studies to help better understand the topic.

## 4. Conclusions

While P-use efficiency is low for potato production in Northeast Florida with a long history of 130 plus years, the soil contains plenty of Mehlich-3 extractable phosphorus. This study was conducted with chipping potato 'Atlantic' to evaluate the effects of P fertilization rates on tuber yield and quality on a large-scale (the total trial area was 13.26 hectares per year). Two local commercial potato farms participated in this study for

three consecutive years. The soils on the two farms were not only rich in Mehlich-3 extractable P but also in active metals such as Al and Fe. The Mehlich-3 soil extractable P was 497 and 220 mg kg$^{-1}$ P at Farms A and B. The results of this study showed 48.9 kg P ha$^{-1}$ P still significantly increased tuber yield as compared with the control with 12.2 kg P ha$^{-1}$ fertilizer P. By applying 48.9 kg P ha$^{-1}$ fertilizer P, the economic return was \$312 ha$^{-1}$ ranging from \$195 to \$484 ha$^{-1}$ for Farm A. For Farm B, the economic return was \$757 ha$^{-1}$ ranging from \$355 to \$1,410 ha$^{-1}$ for Farm B. The external tuber quality was significantly better with 48.9 kg P ha$^{-1}$ fertilizer P than with the control with 12.2 kg P ha$^{-1}$ fertilizer P. During each of the three growing seasons, soil pH declined one pH unit resulting from nitrification of the N fertilizer applied and cation nutrient uptake by the potato plants. The reduction of soil pH activated soil Al and Fe which immobilized both the fertilizer P and soil legacy P and minimized P-use efficiency for potato production in the area. The immobilization and minimization can explain why potato plants grown on the soil with high legacy P still respond to P application positively. To enhance agricultural and environmental sustainability, new fertilization programs are in urgent need to better manage P fertilization for potato production. The findings of this study implied that future studies should focus on the P fertilization method to minimize the chemical fixation of P by soil Al and Fe. Although the soil is abundant in Mehlich-3 extractable P, the amount of approximately 50 kg P ha$^{-1}$ is considered the right application rate of P for sustainable potato production in the area because of the chemical fixation.

**Supplementary Materials:** The following supporting information can be downloaded at: https://www.mdpi.com/article/10.3390/agronomy12040812/s1, Table S1: Air (60 cm) and soil (−10 cm) temperature (Celsius) of Farm A and B from January 2018 to June 2020 (Reference [38] is cited in the supplementary materials), Figure S1: The diagnostic plot of the regression of yield on P rate, year of measure, and farm, Figure S2: The diagnostic plot of the regression of P content on P rate, days after planting in the growing season, farm, soil pH, and nutrient contents before fertilization for the year of 2020, Figure S3: The diagnostic plot of the regression of P content on P rate, days in the growing season, farm, soil pH, and nutrient contents before fertilization for the year of 2019, Figure S4: The diagnostic plot of the regression of P content on P rate, days in the growing season, farm, soil pH, and nutrient contents before fertilization for the year of 2018, Figure S5: The diagnostic plot of the regression of total cull on P rate, farm, and year.

**Author Contributions:** Conceptualization, G.L.; methodology and formal analysis, Y.Q., Z.S. and G.L.; writing—original draft preparation Y.Q.; writing—review and editing Y.Q., T.F., Z.S., F.B., W.M., G.E., D.D., K.M., M.C. and G.L.; finalizing and project administration G.L., and funding acquisition, K.M., D.D., G.L. (25,206) and Z.S. (632,688). All authors have read and agreed to the published version of the manuscript.

**Funding:** This work was financially supported by Florida Department of Agricultural and Consumer Services (FDACS) (Contract ID: 25206) and The Simons Foundation (Contract ID: 632688).

**Institutional Review Board Statement:** Not applicable.

**Informed Consent Statement:** Not applicable.

**Acknowledgments:** Timothy Wilson helped collect soil samples and harvest for all the three growing seasons. Scott Chambers and Christian Christensen helped apply fertilizer and examine external and internal tuber qualities. Bruce Matthew provided truck scale to measure tuber yield for Farm B. Edward Hanlon, Ashok Alva, and Yuncong Li reviewed and improved the manuscript. Scott Parker and Steven Singleton provided lands and potato seeds for the study.

**Conflicts of Interest:** The authors declare no conflict of interest.

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
