# Peer review of "Effect of Phosphorus Fertilization on Yield of Chipping Potato Grown on High Legacy Phosphorus Soil"

_agronomy, doi:10.3390/agronomy12040812_

Round 1

Reviewer 1 Report

Article well designed and worth presentation can be accepted after minor revision. The following points need to be addressed before acceptance.

  1. The abstract must be clear and concise. Reframed it in qualitative form.
  2. The introduction is ok however, the hypothesis and objective must be in line with the defined problem, hence need to be restricted which will improve the readership.
  3. The methodology is ok but the extraction method needs to straighten.
  4. Results and discussion are ok small spelling check suggested.
  5. The conclusion must be shortened and client-oriented.
  6. References need to be cross-checked.

Author Response

  1. The abstract must be clear and concise. Reframed it in qualitative form.

Response: the abstract was reframed in P1, L18-41.

  1. The introduction is ok however, the hypothesis and objective must be in line with the defined problem, hence need to be restricted which will improve the readership.

Response: revised accordingly.

  1. The methodology is ok but the extraction method needs to straighten.

Response: added a few lines accordingly.

  1. Results and discussion are ok small spelling check suggested.

Response: checked accordingly.

  1. The conclusion must be shortened and client-oriented.

Response: rewritten accordingly.

  1. References need to be cross-checked.

Response: checked accordingly.

Reviewer 2 Report

This study estimated the “Effect of Phosphorus Fertilization on Yield of Chipping Potato Grown on High Legacy Phosphorus Soil”. Indeed, influence of fertilization on plant growth and productivity of special interest. There are a lot of comments that should be taken into account by authors, which I believe are significant and important aspects that need to be thoroughly addressed in authors revision.

The main concern is:

Abstract:

(1) The presentation of the fertilization treatments is not clear – please complete it.

(2) The presentation of the results is also not clear.

This section should be carefully and completely revised.

(3) Language used in this section is very bad.

Introduction:

(4) Please illustrate the importance of potato. Please also illustrate its nutritional values and its importance for the consumers.

(5) Please also illustrate by more details and by incorporate the latest references the impacts of using P fertilization on potato growth and productivity.

(6) Research purpose is not clear - please re-write it.

(7) At the end of this section, authors should illustrate what hypothesis this investigation aimed to test.

(8) Moreover, to verify this hypothesis, mention to the measured parameters.

Material and methods:

(9)  Please, give complete soil chemical analysis.

(10) Please, give the full element composition of Mehlich-3 extraction. Did you make the Mehlich-3 extraction or what?

(11) Authors did not mention to the time of fertilizers application. They only mentioned to the amount of applications.

(12) The presentation of the treatments is not clear. This section should be completely revised.

(13) I wonder that the authors have not shown any growth parameters like plant height, shoot and root fresh weights, shoot and root dry weights, leaves area, leaves number, etc.

Discussion:

(14) Authors could discuss the results in their subtitles.

(15) In each paragraph, begin with a sentence that introduces what the paragraph is about, link your findings with those in the literature, and finish with what are the main points. Furthermore, you should avoid repeating the results.  

(16) Authors do not mention some literature on this topic and do not compare their results with those of previously published papers.

(17) Authors should explain more clearly how they represent an advance on previous studies involving plant type (potato), P fertilization, and the interactions between them.

(18) Authors should discuss how their results fill the gap of previous studies.

(19) At the end of this section, all evaluated parameters should be well integrated and discussed.

Conclusion:

(20) This section should be carefully and completely revised.

References:

(21) References used by authors are not the newest one; Please use the newest one.

Linguistic quality:

(22) The English language should be revised.

Author Response

Abstract:

  1. The presentation of the fertilization treatments is not clear – please complete it.

Response: the fertilization treatments were revised in P1, L23-24.

  1. The presentation of the results is also not clear. This section should be carefully and completely revised.

Response: the results was revised in P1, L27-34.

  1. Language used in this section is very bad.

Response: the language used in this section was revised.

Introduction:

  1. Please illustrate the importance of potato. Please also illustrate its nutritional values and its importance for the consumers.

Response: the importance of potato and its nutritional values were added in P1, L39-41.

  1. Please also illustrate by more details and by incorporate the latest references the impacts of using P fertilization on potato growth and productivity.

Response: more details about impacts of using P fertilization on potato growth and productivity were added in P2, L133-138, with the latest references 17 and 18.

  1. Research purpose is not clear – please re-write it.

Response: the research purpose was revised in P2, L145-147.

  1. At the end of this section, authors should illustrate what hypothesis this investigation aimed to test.

Response: the hypothesis was added in P2, L147-149.

  1. Moreover, to verify this hypothesis, mention to the measured parameters.

Response: the measured parameters were mentioned in P2, L149-150.

Material and methods:

  1. Please, give complete soil chemical analysis.

Response: in these acidic soils, we focused on P, Al and Fe measurements in soil elemental analyses.

  1. Please, give the full element composition of Mehlich-3 extraction. Did you make the Mehlich-3 extraction or what?

Response: added.

  1. Authors did not mention to the time of fertilizers application. They only mentioned to the amount of applications.

Response: we put the information of fertilizer applications in 2.2 Farm Management and Table 2 as well.

  1. The presentation of the treatments is not clear. This section should be completely revised.

Response: revised in 2.1 Experimental Design.

  1. I wonder that the authors have not shown any growth parameters like plant height, shoot and root fresh weights, shoot and root dry weights, leaves area, leaves number, etc.

Response: for these large-scale trials on commercial potato farms, we focused on tuber yields and P, Al, and Fe in these acidic soils and didn’t have these growth parameters.

Discussion:

  1. Authors could discuss the results in their subtitles.

Response: discussion and result section were combined.

  1. In each paragraph, begin with a sentence that introduces what the paragraph is about, link your findings with those in the literature, and finish with what are the main points. Furthermore, you should avoid repeating the results.

Response: all paragraphs in this section were revised and reorganized accordingly from P7 to P14.

  1. Authors do not mention some literature on this topic and do not compare their results with those of previously published papers.

Response: literature on potato tuber quality response to P fertilization was added in P11, L516-520, and P12, L570-573. The comparison between previously published papers and the results were added in P11, L520-521, and P11, 534-537.

  1. Authors should explain more clearly how they represent an advance on previous studies involving plant type (potato), P fertilization, and the interactions between them.

Response: the interactions and differences between this study to the previous work were discussed in P14, L655-659.

  1. Authors should discuss how their results fill the gap of previous studies.

Response: the knowledge gap of previous studies and the significance of this work as well as implications for future research were discussed in P14, L655-661.

  1. At the end of this section, all evaluated parameters should be well integrated and discussed.

Response: the summary of all evaluated parameters was presented in P14, L646-654.

Conclusion:

  1. This section should be carefully and completely revised.

Response: rewritten accordingly.

References:

  1. References used by authors are not the newest one; Please use the newest one.

Response: some newest references (17, 18, 37) were added.

Linguistic quality:

  1. The English language should be revised.

Response: the English language was revised.

Reviewer 3 Report

The experiment conducted by the authors broadens the knowledge on the influence of phosphorus fertilisation on potato yield. The article is generally interestingly written. However, I have some issues that need to be solved by the authors:

  1. Please specify the program you used to perform the statistical analyses.
  2. In the materials and methods section, please include a description of the meteorological conditions in the studied years. For example, what was the precipitation like? Were they similar in particular years? Could they have influenced the results? Please elaborate significantly on this topic in the article.
  3. It is unclear what the water conditions are in the individual plots. How much water was delivered by irrigation? Have groundwater levels been monitored? Do the two farms have similar water relations (soil moisture, water table, etc.) and can the obtained results be compared? This problem has to be solved in the text of the paper.
  4. The discussion in the paper should be developed. Please try to relate your research results to the world literature.
  5. The conclusion section must be rewritten. Authors should include specific results of their research, which extend the current state of knowledge. I suggest including the most important conclusions in a bullet point list.

Author Response

  1. Please specify the program you used to perform the statistical analyses.

Response: specified. R statistical software, version 4.0.2 (R Core Team, 2020)

  1. In the materials and methods section, please include a description of the meteorological conditions in the studied years. For example, what was the precipitation like? Were they similar in particular years? Could they have influenced the results? Please elaborate significantly on this topic in the article.

Response: Total rainfall and ET data at both farms were added for each of the three growing seasons. Seepage irrigation was used to keep the water table between 0.46 and 0.61 m.

  1. It is unclear what the water conditions are in the individual plots. How much water was delivered by irrigation? Have groundwater levels been monitored? Do the two farms have similar water relations (soil moisture, water table, etc.) and can the obtained results be compared? This problem has to be solved in the text of the paper.

Response: Seepage irrigation was used to keep the water table between 0.46 to 0.61 m but we didn’t use flowmeters to monitor the water usage because all the treatments used the same amount. All irrigation water from the same source and there was a fall row per 16 rows for irrigation purpose. This information is added to the 2.1 section.

  1. The discussion in the paper should be developed. Please try to relate your research results to the world literature.

Response: more discussion was added with additional references.

  1. The conclusion section must be rewritten. Authors should include specific results of their research, which extend the current state of knowledge. I suggest including the most important conclusions in a bullet point list.

Response: The conclusion was rewritten.

Round 2

Reviewer 2 Report

Please, find the comments in pdf file.

Author Response

Point-by-point response

Comment 1: first sentence should be revised.

Response: Revised as follows:

Potato (Solanum tuberosum L.) has low phosphorus (P) use efficiency as compared with other vegetable crops.

Comment 2: This study was conducted at two commercial chipping potato farms in Northeast Florida to evaluate different P rates for potato production.

Response: Revised as follows:

This study was conducted at two commercial chipping potato farms (A & B) in Northeast Florida to evaluate different P rates for potato production.

 Comment 3: please remove this part.

Response: removed.

Comment 4: this sentence should be revised.

Response: revised as follows:

The tuber yield data show that potato plants grown on soil with high legacy P still require approximately 50 kg ha-1 P application. This high P requirement is resulted from the combination of high concentrations of active mentals (Al and Fe) and a decrease in pH of one unit in the growing season.

Comment 5: please revise this sentence.

Response: Deleted but added this sentence before the objectives:

P fertilization will significantly increase tuber yield of chipping potato plants grown on soil with high-legacy P due to high concentrations of active Al and Fe and pH decreases in the growing season.

Comment 6: Even you focused on P, Al and Fe measurements, you should give complete soil chemical analysis.

Response: The following lines were added in 2.3.

Before planting, soil samples were collected for soil chemical analysis of either farm in spring 2018. Mehlich-3 extractable P, K, Ca, and Mg were 182±19, 104±12, 1,140±169, and 152±34 mg ha-1 at Farm A and 120±22, 37±15, 742±113, and 65±17 mg ha-1 at Farm B. The CEC was 6.3±0.2 meg (100 g)-1 at Farm A and 3.6±0.2 meg (100 g)-1 at Farm B.